# Assessment of Laying Hens’ Thermal Comfort Using Sound Technology

**DOI:** 10.3390/s20020473

**Published:** 2020-01-14

**Authors:** Xiaodong Du, Lenn Carpentier, Guanghui Teng, Mulin Liu, Chaoyuan Wang, Tomas Norton

**Affiliations:** 1College of Water Resources & Civil Engineering, China Agricultural University, Beijing 100083, China; duxiaodong@cau.edu.cn (X.D.); b20193090700@cau.edu.cn (M.L.); gotowchy@cau.edu.cn (C.W.); 2Division Measure, Model & Mange Bioresponses, Department of Biosystems, KU Leuven, Kasteelpark Arenberg 30, 3001 Heverlee, Belgium; lenn.carpentier@kuleuven.be

**Keywords:** animal vocalisation, THI, SVM, laying hens, animal welfare

## Abstract

Heat stress is one of the most important environmental stressors facing poultry production and welfare worldwide. The detrimental effects of heat stress on poultry range from reduced growth and egg production to impaired health. Animal vocalisations are associated with different animal responses and can be used as useful indicators of the state of animal welfare. It is already known that specific chicken vocalisations such as alarm, squawk, and gakel calls are correlated with stressful events, and therefore, could be used as stress indicators in poultry monitoring systems. In this study, we focused on developing a hen vocalisation detection method based on machine learning to assess their thermal comfort condition. For extraction of the vocalisations, nine source-filter theory related temporal and spectral features were chosen, and a support vector machine (SVM) based classifier was developed. As a result, the classification performance of the optimal SVM model was 95.1 ± 4.3% (the sensitivity parameter) and 97.6 ± 1.9% (the precision parameter). Based on the developed algorithm, the study illustrated that a significant correlation existed between specific vocalisations (alarm and squawk call) and thermal comfort indices (temperature-humidity index, THI) (alarm-THI, R = −0.414, P = 0.01; squawk-THI, R = 0.594, P = 0.01). This work represents the first step towards the further development of technology to monitor flock vocalisations with the intent of providing producers an additional tool to help them actively manage the welfare of their flock.

## 1. Introduction

Animal vocalisations are a fundamental component of animal behaviour, and can be used as useful indicators of the animal welfare state [1,2,3]. Heat stress is one of the most critical environmental stressors in poultry production worldwide [4]. The detrimental effects of heat stress on poultry range from reduced growth and egg production to decreased egg quality and safety [5,6]. Moreover, high ambient temperatures have marked impacts on the behaviour, feed and water intake, heat production, and physiological responses (body temperature, respiratory rate and heart rate) of poultry [7,8], which might elicit specific vocalisation such as alarm, squawk, and gakel calls [9].

The gakel-call, as an indicator of frustration, can serve as an additional indicator of welfare in laying hens [10]. The number of gakel-calls and alarm-cackles can be regarded as potential indicators of frustration when recorded continuously [9]. In addition, alarm-calls were found as indicators of anxiety [1]. Total vocalisation (distress call) rate (i.e., the sum of all calls per animal/unit of time) in chickens is positively correlated with the occurrence of negative or stressful events and an increase in distress calls is considered an indication of compromised animal welfare [11,12]. Therefore, squawk and alarm calls are recognised as essential vocalisations from an animal welfare perspective [13,14]. Counting the number of vocalisations without specifying their types in most cases is not sufficient for welfare assessment [2]. However, little is still known about whether and how these specific vocalisations of laying hens are related to their thermal environment.

The thermal environment of different production systems can be compared using various thermal comfort indices. Among them, the temperature-humidity index (THI) is an important indicator that combines in a single parameter of environmental conditions (temperature, humidity) and provides an idea about the overall effect of the current environment on the tested animals [15,16]. Therefore, this study concentrates on quantitatively measuring frustration-related vocalisations (alarm, squawk, and gakel calls) to explore whether they are useful indicators for evaluating animal thermal comfort condition. This study will lead to a deeper understanding of meaning and relevance with respect to welfare.

Nowadays, modern technology makes it possible to use cameras, microphones, and sensors in proximity to animals to take the place of farmers’ eyes and ears in monitoring animal health and welfare effectively [17,18]. Bio-acoustical-based tools and methods for farming environments are non-invasive techniques for welfare judgements that are recognised as having an essential position in health and welfare monitoring technologies [2]. In animal vocalisation studies, the focus has been to utilise the data for research purposes only. Information on specific vocalisations is often extracted manually from its spectrogram and the intuition of the researcher often drives the choice of parameters. This manual extraction makes the process unsuitable for online and real-time large data analysis [19].

To improve the current state-of-the-art developments, two key components of algorithm development are required for detecting animal vocalisations, i.e., feature extraction and classification. For feature extraction, the source-filter theory of vocal production is a robust framework for studying animal vocal communication. According to this theory, the source-related vocal features (the fundamental frequency, f0) and filter-related features (formants, F1–F3) are considered as valid feature input parameters [20]. For example, filter-related features can help in decoding the differences between estrus and feed anticipating vocalizations in cows [21]. Both source-related and filter-related vocal features are essential in discriminating individual animal vocalisation [22]. For the classification of the vocalisations, another algorithm is necessary to operate on the feature output. Classification through support vector machines (SVM) has been shown to give excellent results in animal sound recognition [23,24,25]. Although SVM was initially designed for binary (2-classes) classification [26], later research has extended it to multi-category regression and classification research, making it suitable for classifying vocalisations [27]. For example, the sound of healthy and avian-influenza-infected chickens in the laboratory can be classified using SVM with an accuracy rate from 84% to 90% [28]. Livestock vocalisation classification algorithms based on SVM have been developed successfully, where they are targeting livestock-related sound with high accuracy for data sets from different species (sheep: 99.29%, cattle: 95.78%, dogs: 99.67%) [29].

Given the above rationale, this study aims at developing a sound-based monitoring method to automatically evaluate laying hens’ thermal comfort through automated vocalisation detection. A key innovation of this study is the application of sound techniques to automatically and remotely assess laying hens’ heat stress conditions for the first time. The objectives are as following: (i) automatic hens’ call detection, (ii) sound recognition model, (iii) application of the algorithm.

## 2. Materials and Methods

### 2.1. Animal and House

Experiments were carried out at a pilot farm (the Shangzhuang experimental station of China Agricultural University, Beijing, China). There, 100 hens (breed: Jingfen) were reared in a perch husbandry system at the age of 18 to 20 weeks (Figure 1). The area of this system was 4.5 m L × 1.6 m W × 2.8 m H (Figure 2). There was ad libitum access to food and water and a timer-controlled light schedule (light period: 8:00 a.m. to 6:00 p.m.) was conducted during the experimental period. Experiments were performed inside a temperature-controlled chamber to monitor and record hens’ vocalisation in different thermal environments (THI level). THI was recorded every five minutes and divided into four levels: comfort zone (THI < 70), alert zone (THI 70–75), danger zone (THI 76–81) and emergency zone (THI > 81) [15]. Additionally, sounds were classified into four categories: gakel, alarm, squawk, and others (Table 1). All experimental procedures were conducted in conformity with the Jingfen management guidelines and treatments for the care and use of laboratory animals. All recording procedures were non-invasive and did not cause any disturbance to the animals during their regular daily activity.

### 2.2. Data Collection

The sound was recorded using a top-view Kinect V1 (a sound monitoring system) for Windows (Microsoft Corp., Redmond, WA, USA). The system was installed 3.0 m above the floor. Sound data were acquired continuously in Waveform Audio File Format (1 channel, 32-bit resolution, 16,000 Hz, recording at approximately 55 s of each file). Frequency information up to 8000 Hz (= Nyquist frequency) was considered enough to automatically detect the sound events of interest at a low computational cost [31]. The Kinect was connected to a mini industrial personal computer (IPC) via a USB cable (Figure 1). A mobile hard disk drive (HDD) with 2 TB storage and USB 3.0 was used to record sound data. During this experiment, 105 h of data were recorded. The original data are uncompressed, so more than 23 GB of data were stored on the HDD.

### 2.3. Sound Signal Pre-Processing and Labelling

The sound signal was divided into frames before pre-processing. Each frame consisted of N = 512 samples to comply with the size of the discrete Fourier transform (DFT). Overlapping frames with a 50% overlap were recommended to avoid losing information, and the Hamming window was used to reduce edge effects and spectral leakage in each frame. Then, each frame was filtered based on spectral subtraction and the band-pass filter algorithm in MATLAB [32]. After filtering, sound data was labelled by manual audio-visual inspection [33]. Audacity^®^ software version 2.3.0 was used to replay and label the data by two trained technicians to inspect each recording and annotate the start and end time of all call events. Meanwhile, there were also some dubious labels after validated by the animal expert. These labels were scored by the following method to decide which one should be removed or could be added as other call types in Table 1.

In addition, manual labelling was used to score the quality of measured data corresponding to 3 ratings [34]. Overlapped sounds labelled as rating 1 were not selected as training and testing datasets due to their complex acoustic features and limitations in sound source separation technology. Moreover, unclear sounds labelled as rating 2 were not regarded as training or testing datasets. Only clear vocalisations in rating 3 were used for data analysis in the following section.

### 2.4. Algorithm for Automatic Hens’ Call Detection

The developed automatic detection algorithm could be split into three parts: automatic sound event selection, feature extraction, and classification (Figure 3). In the pre-processing step, the raw sound data were cleaned by removing background noise. Random 8 h raw data were selected for manual labelling and another 21.5 h data were used for automatic sound event detection to annotate the beginning and end of possible hen calls. Next, each frame-based feature vectors were calculated and then they were classified into different types of calls. The number of labelled sounds for training and testing were 2368 and 228,200, respectively. The entire algorithm was developed in NI LabVIEW 2015 (American National Instrument Corp., Austin, TX, USA) and MATLAB R2012a (MathWorks Inc., Natick, MA, USA).

#### 2.4.1. Automatic Sound Event Selection

The objective was to detect the calls of hens. A good selection of each sound event such as the exact beginning and end, will improve the success rate of the classification algorithm [34]. Filtered hen calls contain more energy than the background noise, so the envelope of energy method was selected to perform the sound detection. This automatic sound event selection method extracted endpoints from a continuous recording based on a manually selected threshold (0.35 in this trial) [34,35]. The algorithm was gradually optimised by maximising the overlap factor between the algorithm output and the labelled sound [36].

#### 2.4.2. Feature Extraction

After testing 69 spectral and temporal features, only nine features were chosen according to their observable and distinguishing difference among four types of calls shown in their feature histograms (Table 2). These features were competent enough for adequate classification. The equation for spectral energy is expressed as:(1)$$E=\log_{10}\left({\sum_{n=1}^{L}\left(\frac{X\left(n\right)}{X_{\max}}\right)}^{2}\right)$$
where *n* = 1, …, *L* represents the index of the vector of frequency bins, and *X*(*n*) is the energy of the *n*th frequency bin. First, the spectral energy is normalised by dividing the maximal energy of all frequency bins.

#### 2.4.3. Classification

The SVM classifiers were developed based on statistical learning theory [38]. They are widely used because of their generalization ability. The basic idea is to transform input vectors into a high-dimensional feature space using a nonlinear transformation function, and then to do a linear separation in the feature space. The SVM algorithm can construct a variety of learning machines by use of different kernel functions [23]. Three kinds of kernel functions are usually calculated as:Polynomial kernel function:(2)$$K(x,y)\;=\;{\left(gamma*x^{T}y+coef0\right)}^{\mathrm{degree}}$$Radial basis function with Gaussian kernel:(3)$$K(x,y)=\exp\left(-gamma*{\left|x-y\right|}^{2}\right)$$Sigmoid function:(4)$$K(x,y)=\tanh\left(gamma*x^{T}y+coef0\right)$$

Although they were initially designed to solve two-class classification problems, multi-class classifications can be performed using the two common methods: one-vs.-all and one-vs.-one [39]. The former method was applied in the LabVIEW program explicitly designed for classifying the hens’ call. The default maximum iterations and tolerance was set at 10,000 and 0.0001, respectively.

A total of 2368 clips were manually labelled for training data and validation data. To avoid overfitting the data, k-fold cross-validation was used to estimate the classification performance [40]. Given the limitation of data size, the sound data were split into five smaller sets and the algorithm was trained using four sets (80% of the data) and validated on the remaining set (20% of the data). The average of the five validation sets was used to calculate the performance of the algorithm.

#### 2.4.4. Performance Estimation

The overlap factor indicates the similarity between the algorithm output and the labelled sound to evaluate the selection performance. Moreover, the performance of both event selection and classification was evaluated by calculating two different statistical measurements, i.e., sensitivity (recall) and precision [36].
(5)$$\mathrm{sensitivity}=\frac{\mathrm{number}\;\mathrm{of}\;\mathrm{true}\;\mathrm{positives}}{\mathrm{number}\;\mathrm{of}\;\mathrm{true}\;\mathrm{positives}\;+\;\mathrm{number}\;\mathrm{of}\;\mathrm{false}\;\mathrm{negatives}}\times 100\%$$
(6)$$precision=\frac{number\;of\;true\;positives}{number\;of\;true\;positives+number\;of\;false\;positives}\times 100\%$$

## 3. Results and Discussions

### 3.1. Algorithm Performance

To obtain optimised parameters for the training SVM model, the code: classification search parameters.vi was implemented on the LabVIEW software platform. The optimal hyper-parameters of the SVM model are C_SVC for SVM type, polynomial for kernel type, value 1 for C (penalty factor), value 3 for the degree, value 0.2 for gamma, and value 1 for coef0. Table 3 presents the final confusion matrix in the SVM training and testing stage, where 5-fold cross-validation was used to estimate the classification performance. As shown in Table 3, gakel calls can be easily classified as other calls, which results in the lowest sensitivity. The leading cause of the low recognition rate might be a relatively small number of gakel calls. With an improved training set, classification success would increase. In terms of the precision parameter, all the call types, except for alarm call, can be easily classified as an alarm call. In this experiment, an alarm call and squawk call could be more easily confused with each other. The stage of the manual label and automatic sound event selection was significant to the final classification performance.

As shown in Table 4, the alarm call shares the highest sensitivity rate and the gakel call shares the highest precision rate. Although the gakel call shares the lowest sensitivity, the method has a high recognition rate with reference to the overall call. The average parameters are both above 95.0%.

Specific vocalisations such as alarm, squawk, and gakel calls in chicken can be regarded as a sound indicator of health and/or well-being. In this paper, a bio-acoustical method was developed to automatically recognise these specific vocalisations, which is better than manual observation using subjective and time-consuming judgement [10,13]. Moreover, the method can realise automatic hen call detection in a large data set. However, it is impossible to determine by manual examination of the spectrograms or by subjectively listening to a huge volume of recordings. In this study, the SVM model has shown excellent results in animal sound recognition. The optimal parameters of the SVM are determined through 5-fold cross-validation, which can easily overcome the problem of overfitting or overlearning in the training set. The resulting performances were 95.1 ± 4.3 (for sensitivity rate) and 97.6 ± 1.9 (for precision rate). Other similar sound recognition performances were 92.1% for accuracy rate [23], 92% (an average accuracy for three behaviours), and 84% (an average precision for three behaviours) [25], 91.9–94.3% (frog calls) [41], 88.75% (accuracy of heat stress condition in turkeys) [42], 95.78–99.67% (three animal vocalisations: sheep, cattle, and dog) [29], and 66.7% for sensitivity rate and 88.4% for precision rate (sneeze or no-sneeze) [36], respectively. Compared to other references, the number of extracted features was less than that in other methods. For example, chicken vocalisation was analysed in the time domain, and 25 statistical features were extracted from the sound signals and the five best features were selected using the FDA (Fisher Discriminate Analysis) method to obtain features as the classifier ANN’s (artificial neural network) input parameters [27]. In this study, only nine features were extracted and the five best features were sufficient to achieve a high precision rate of 97.0% and a high sensitivity rate of 94.7% (Figure 4 and Figure 5). In contrast, the reference could only achieve an average accuracy of 88.68% (train) and 83.33% (test) [27].

As shown in Figure 4 and Figure 5, sequential feature selection was performed to easily observe their contribution to the recognition rate among different features. Although the maximum precision rate was nine features, the first four features (feature order 1, 8, 7, and 2) had already achieved a stable and high recognition rate. The same was true of the sensitivity rate, while the difference was that five features achieved a stable and high recognition rate. The corresponding selected features were feature order 7, 9, 5, 1, and 8.

### 3.2. Thermal Comfort Assessment

After the development of the sound recognition model, this study tried to assess and analyse the hens’ thermal comfort condition using recognised flock calls. The utility of this model in monitoring animal heat stress was based on a significant correlation between THI and the number of calls. As shown in Table 5 and Table 6, the Pearson correlation was significant at the 0.01 level (2-tailed). Based on this correlation, in the next step, it was possible to develop a vocal-based algorithm to predict animal thermal comfort levels automatically.

Using the auto-label sound data in the testing dataset and SVM classifier, we could realise the quantitative analysis of all vocalisations under different thermal conditions. Figure 6 presents the difference between the number of classified hen calls under different thermal conditions. We found that the number of levels of alarm calls presented more than that of squawk calls produced in the alert zone. In contrast, the latter one was higher than that of the alarm calls produced in the emergency zone. It could be concluded that an extreme thermal environment may easily cause more distress calls than that in a thermal comfort environment. There may be several explanations as follows. Squawk and alarm calls are both recognised as essential vocalisations from an animal welfare perspective [9]. The squawk call is a vocalisation of high frequency that can be made when the bird is startled or is experiencing pain [14]. Meanwhile, the alarm call is a rather soft turmoil reaction that occurs in response to the appearance of animals or people, things, and mild external changes [13]. Another research found that chickens under thermal distress conditions produced more distress calls than those under thermal comfort conditions [43]. Squawk calls are consistent with the results of this reference. However, alarm calls show the opposite pattern, which could not be explained in this study.

Hens tend to make a cautionary sound in response to changes in the external stimuli and respond sensitively and together as a group to the abrupt change of such sounds [14]. Specific vocalisations such as alarm, squawk in the chickens are correlated significantly with THI, which is meaningful and was observed in this chicken research. These sounds could be regarded as sound indicators for estimating the thermal comfort of chickens in different thermal environments. In some cases, vocalisations are not always presented. Chronic stress and even chronic pain seem to evoke no vocalisation in most animals [44].

Moreover, chronically adverse physical conditions may be expressed by an irregular voiceprint of normal vocalisation [44]. Therefore, further research will study changes in the detailed voiceprint of specific vocalisations under chronic and acute stress conditions. The work conducted in this study was aimed at minimising the challenges to those expected in commercial deployment of the technology. With the significant correlation between THI and specific vocalisations, farmers can only use a single sound indicator to assess bird thermal comfort levels without collecting environmental parameters automatically. As shown in Figure 7, the method has the potential for online farm application using fewer sensors and perceiving individual animal behaviour rather than the ambient environment. Also, the system could realise anomalous sound monitoring and thermal comfort evaluation.

Additionally, there have been several attempts to measure heat stress responses in poultry. The noise analysis method has demonstrated that it can evaluate thermal comfort [43]. However, the research only realised the interpretation of the noise amplitude and noise frequency spectrum of bird vocalisations under thermal distress conditions using manual statistic instead of automatic detection. Another automatic stress recognition system using sound techniques was designed to detect laying hens physical stress caused by changes in the temperature and mental stress from fear [45]. The average accuracy was 96.2% based on the SVM binary-classifier. Meanwhile, the target sound is none instead of the specific vocalisations, which may be difficult for manual observation or application of automatic sound detection in commercial farms.

Despite ambient environment monitoring sensors, the possibility still exists for situations that cause laying hens stress. Unfortunately, since we were only able to monitor and stress one flock and record the birds’ specific vocalisations, more data should be gathered in different settings. It should be verified that the method worked in this henhouse and could be generalised to other settings. Another essential and ill-considered factor in this experiment was the stocking density because it has been proven that as group size decreased, the number of distress calls increased [11]. In future research, a larger sample size would help us detect the subtle differences in the number of levels of specific vocalisations under thermal conditions. Moreover, specific vocalisations correlated with non-welfare animal behaviour and other stressors such as pollutants, harmful gas, and equipment failure would be a valuable area of research.

## 4. Conclusions

We proposed an innovative way of assessing thermal comfort, correlating THI with chicken vocalizations. The method can realise automatic hen call detection and classification. The method yielded a precision rate of 97.6 ± 1.9% and a sensitivity rate of 95.1 ± 4.3%. Based on this algorithm, the thermal comfort level of birds may be evaluated easily. Owing to thermal inertia in a commercial henhouse, this method might also be used as an early warning detection tool to avoid the lag of monitoring the ambient environment. Moreover, we found that specific vocalisations such as alarm and squawk calls of chickens were correlated significantly with heat stress indices (THI). Finally, the bio-acoustical method with data that accumulated through the monitoring system may be considered as a stressor indicator to evaluate animal welfare in the future.

## Figures and Tables

**Figure 1 sensors-20-00473-f001:**
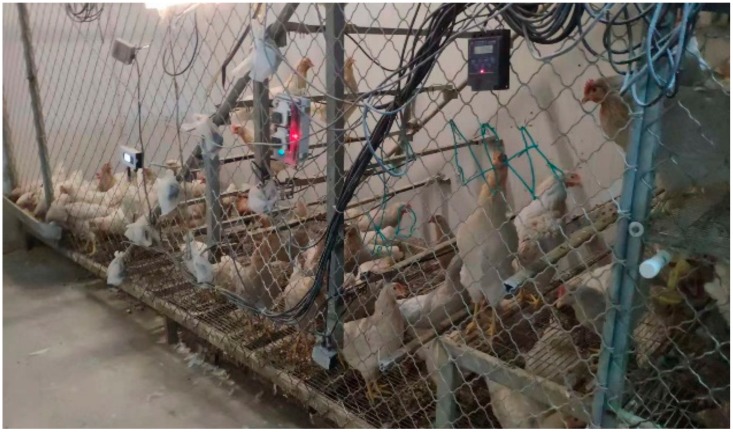
On-site perch husbandry system.

**Figure 2 sensors-20-00473-f002:**
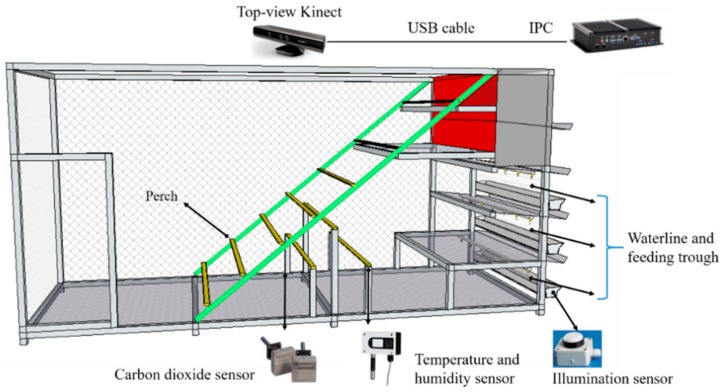
Schematic of the experiment platform.

**Figure 3 sensors-20-00473-f003:**
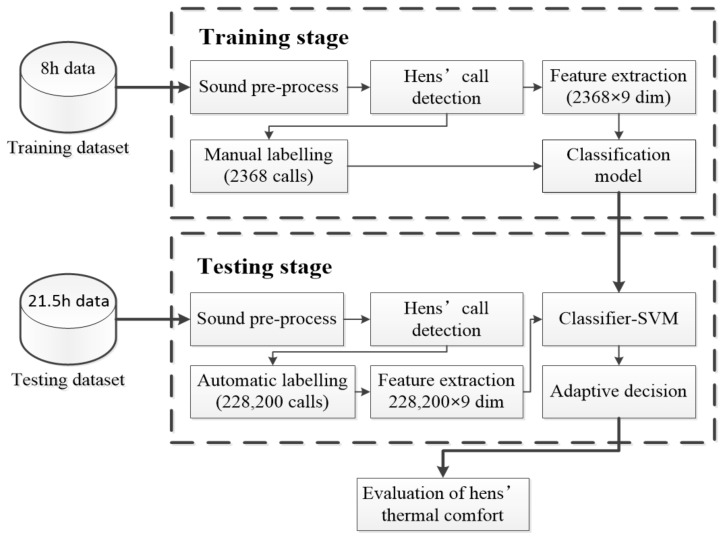
Flow chart of automatic hens’ call detection.

**Figure 4 sensors-20-00473-f004:**
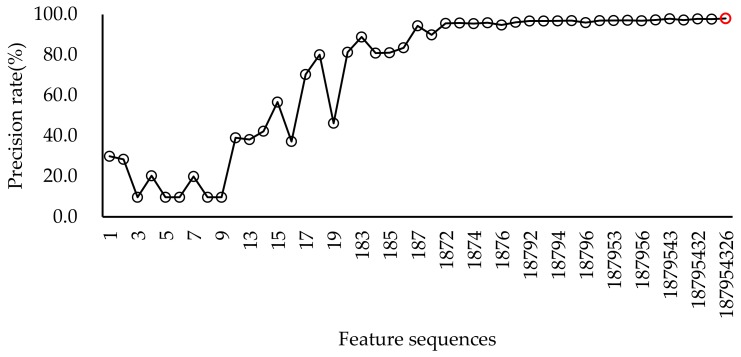
The plot of precision rate in different feature sequential selection. The red circle marks the maximum recognition rate.

**Figure 5 sensors-20-00473-f005:**
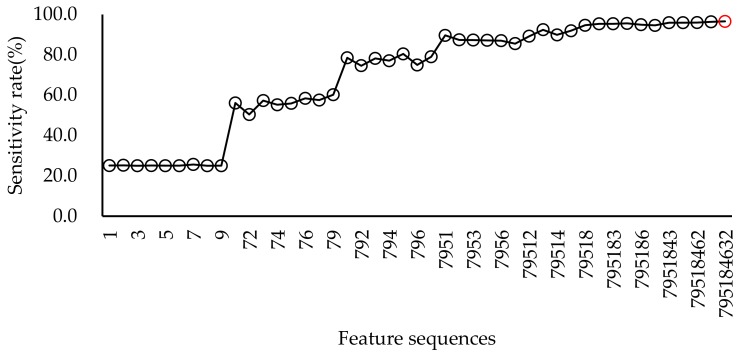
The plot of sensitivity rate in different feature sequential selection. The red circle marks the maximum recognition rate.

**Figure 6 sensors-20-00473-f006:**
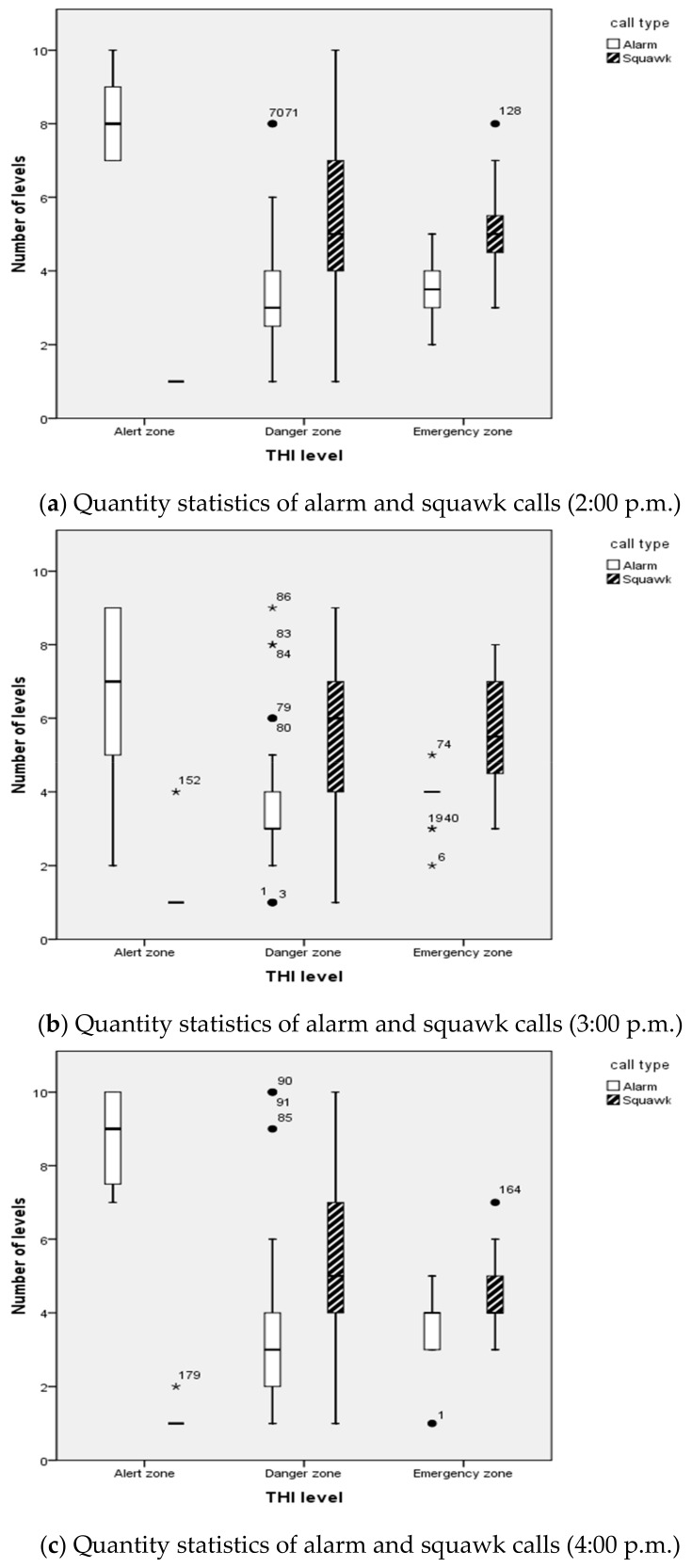
The number of levels of hen calls in different thermal environments. Alert zone (THI 70–75), danger zone (THI 76–81), and emergency zone (THI > 81). The number of levels axis indicates the normalised range (from level 1 to level 10) of call quantity per five minutes.

**Figure 7 sensors-20-00473-f007:**
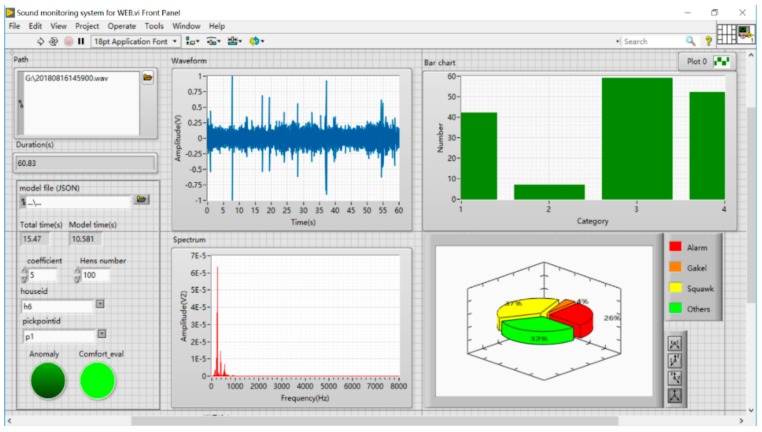
LabVIEW panel of the sound monitoring system for WEB.

**Table 1 sensors-20-00473-t001:** Description of different hens’ vocalisations.

Call Type	Description
Gakel	Soft, brief (<0.2 s) repetitive notes generally with a wide frequency range; often emphasize low frequencies (below 2 kHz). Notes with definite, clear harmonic structure [10,30].
Alarm	High pitched sound of duration (<0.2 s) with a distinct harmonic structure, moderately loud (similar to alert calls [9,13,30]).
Squawk	Component notes are short (<0.1 s) with an abrupt onset and ending and cover a wide frequency range. This call is a moderately loud sound (similar to distress cries [9,13,30]).
Others	Other hens’ vocalisations. Total vocalisation rate is positively correlated with event aversiveness in domestic chickens [13].

**Table 2 sensors-20-00473-t002:** Description of feature parameters.

Feature Parameters	Description	Order
Jitter_f0	Mean absolute difference between frequencies of consecutive f0 periods divided by the mean frequency of f0 (fundamental frequency) [20]	1
Jitter_F1	Mean absolute difference between frequencies of consecutive F1 periods divided by the mean frequency of F1 (the first formant) [20]	2
Jitter_F2	Mean absolute difference between frequencies of consecutive F2 periods divided by the mean frequency of F2 (the second formant) [20]	3
Shimmer_F1	Mean absolute difference between the amplitudes of consecutive F1 periods divided by the mean amplitude of F1 [20]	4
Shimmer_F3	Mean absolute difference between the amplitudes of consecutive F3 periods divided by the mean amplitude of F3 (the third formant) [20]	5
ZCR	The zero-crossing rate (ZCR) of an audio frame is the rate of sign-changes of the signal during the frame [37]	6
Spectral spread	The spectral spread is the second central moment of the spectrum [36]	7
Spectral energy	Refer to Equation (1)	8
Spectral centroid	The spectral centroid is the centre of ‘gravity’ of the spectrum [36]	9

**Table 3 sensors-20-00473-t003:** Confusion matrix of support vector machine (SVM) modelling.

Real Call Type	Classified by 9 Features					
	Alarm	Gakel	Squawk	Others	Total	Sensitivity (%)
Alarm	906	0	5	6	917	98.8
Gakel	7	96	3	4	110	87.3
Squawk	18	0	727	5	750	96.9
Others	8	0	13	570	591	96.4
Total	939	96	748	585	2368	-
Precision (%)	96.5	100.0	97.2	97.4	-	-

Note: - denotes null value.

**Table 4 sensors-20-00473-t004:** Classification performance of the SVM model.

Call Type	Classification Performance	
	Sensitivity ± SD (%)	Precision ± SD (%)
Alarm	98.4 ± 0.5	95.5 ± 1.4
Gakel	88.9 ± 1.4	100.0 ± 0.0
Squawk	96.1 ± 1.7	96.6 ± 0.4
Others	97.0 ± 1.3	98.1 ± 0.5
Total	95.1 ± 4.3	97.6 ± 1.9

**Table 5 sensors-20-00473-t005:** Correlations between the alarm and temperature-humidity index (THI).

		THI	Alarm
THI	Pearson Correlation	1	−0.414 **
	Sig. (2-tailed)		0.008
	N	40	40
Alarm	Pearson Correlation	−0.414 **	1
	Sig. (2-tailed)	0.008	
	N	40	40

** Correlation is significant at the 0.01 level (2-tailed).

**Table 6 sensors-20-00473-t006:** Correlations between the squawk and THI.

		THI	Squawk
THI	Pearson Correlation	1	0.594 **
	Sig. (2-tailed)		0.000
	N	40	40
Squawk	Pearson Correlation	0.594 **	1
	Sig. (2-tailed)	0.000	
	N	40	40

** Correlation is significant at the 0.01 level (2-tailed). 0.000 denotes that the value is less than 0.001.

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
