# Peer review of "Assessment of Laying Hens’ Thermal Comfort Using Sound Technology"

_sensors, 2020, doi:10.3390/s20020473_

Round 1

Reviewer 1 Report

Sensors laying hen sound

This is an interesting paper that sets the foundation for using chicken vocalizations to monitor hen welfare.  While the model used here is heat stress, the authors should acknowledge and consider investigating the utility of this model in monitoring other types of stressors, as different stressors may elicit different responses.

Line 23: you cannot “prove” a hypothesis.  You can reject or fail to reject.  I believe you meant “illustrated”

Food for thought: the sqwaks and gakel calls are typically in response to immediate and acute stressors (e.g., surprise, fear, rage).  Most heat stress events will occur over a longer period of time and may therefore be less noticeable as an immediate and acute threat.  What role do the authors thinks this plays in the usefulness of these vocalizations in detecting a chronic or subclinical onset of heat stress.  Would monitoring different types of vocalizations be more beneficial for detecting heat stress while others better for being indicative of an immediate welfare challenge (e.g., hunger, thirst).  Taking some time to explain this would be beneficial to the argument.  What vocalizations would be indicative of chronic stress versus acute – because once heat stress starts, it will take some time for the animals to thermoregulate and the stressor to be alleviated.  I challenge the authors to include this perspective in their development of algorithms and interpretation of output. 

Line 23-25: I believe the authors mean that as THI increases, the frequency of squawks and alarm calls increases accordingly (R = x.xx, P = x.xx). 

Line 25-26: can be deleted

Line 26-27: overstated and would benefit from specificity.  The results support the further development of technology to monitor flock vocalizations with the intent of providing producers an additional tool to help them actively manage the welfare of their flock.

Line 33: revise “…indicators of the animal welfare state”

Line 43: “chickens”

Line 45: suggest replace “impairing” with “compromised”

Lines 123-126:  I find this difficult to believe.  After watching thousands of hours of chicken video, it is difficult to determine whether specific individuals are vocalizing in a group.  What was the validation strategy?  How many observers were there?  What was their inter-rater reliability?

Section 3.2: I think there is something missing here.  Were the vocalizations combined and then regressed?  More explanation of the analysis is needed here.

Line 276-277: revise for grammar

What influence does the automatic machinery in a hen house have on the ability to apply this technology to commercial production?  The frequency of machinery that is used to care for the hens may mask some of the vocalizations needed to monitor welfare.  Something to consider as you are identifying optimal ranges for vocalization detection.  There also may be breed and age differences in which vocalizations are informative to producers.

Reviewer 2 Report

This paper describes a method to detect and distinguish chicken vocalizations using machine learning methods, as well as using the detected vocalization events to estimate the thermal comfort of the chicken. 

Overall the paper is well written, with most of the technical details clearly described. The reviewer only has a few comments / questions:

Line 118: Is the choice of 512 sample FFT optimal? The time/frequency resolution (which depends on the choice of FFT size) may have a fairly high impact on the vocalization detector's performance, and it may also affect the result of the classifier. The authors could consider trying with various FFT size and see if the results can be improved. Line 130: The reviewer wonders if excluding the less clear sounds from the training data of the SVM may negatively impact the accuracy of the training, especially since these vocalizations naturally occur in the experiment environment. Has the authors tried to check the failures in the confusion matrix, to see if the false detections are among the less clearly recorded vocalizations? Line 162: What does it mean by "consecutive f0 periods"? It would be much easier to understand these metrics/features if the equations were provided, just like Eq.1. Line 254: The authors should consider elaborating on the definition of "number of levels", as its meaning does not seem very clear. Is this the SPL level of the vocalization? Should the unit be normalized by time, such as "number of levels per hour"?
